# Simulation of the Refractive Index Variation and Validation of the Form Deviation in Precisely Molded Chalcogenide Glass Lenses (IRG 26) Considering the Stress and Structure Relaxation

**DOI:** 10.3390/ma15196756

**Published:** 2022-09-29

**Authors:** Cheng Jiang, Carlos Marin Tovar, Jan-Helge Staasmeyer, Marcel Friedrichs, Tim Grunwald, Thomas Bergs

**Affiliations:** 1Department of Fine Machining and Optics, Fraunhofer Institute for Production Technology IPT, 52074 Aachen, Germany; 2Laboratory for Machine Tools and Production Engineering (WZL), RWTH Aachen University, 52074 Aachen, Germany

**Keywords:** PGM, chalcogenide glass, index drop, FEM, stress relaxation, structure relaxation, ray-tracing, OPD

## Abstract

Precise infrared (IR) optics are core elements of infrared cameras for thermal imaging and night vision applications and can be manufactured directly or using a replicative process. For instance, precision glass molding (PGM) is a replicative manufacturing method that meets the demand of producing precise and accurate glass optics in a cost-efficient manner. However, several iterations in the PGM process are applied to compensate the induced form deviation and the index drop after molding. The finite element method (FEM) is utilized to simulate the thermomechanical process, predicting the optical properties of molded chalcogenide lenses and thus preventing costly iterations. Prior to FEM modelling, self-developed glass characterization methods for the stress and structure relaxation of chalcogenide glass IRG 26 are implemented. Additionally, a ray-tracing method is developed in this work to calculate the optical path difference (OPD) based on the mesh structure results from the FEM simulation. The developed method is validated and conducted during the production of molded lenses.

## 1. Introduction

Precision glass molding (PGM) shows a great advantage in manufacturing aspherical lenses, or even freeform optics, which cannot be easily produced by glass grinding and polishing [1]. This replicative approach is a scalable and economic solution, which keeps the high form accuracy and fine surface quality of the optical component. By this method, a glass preform is loaded in the molding chamber and heated above its transition glass temperature (*T_g_*). Once the glass preform and molding tools have reached a uniform temperature, a molding force is applied to deform the glass preform between the two molding tools. The molded optical component is then cooled by means of a controlled cooling rate and then removed from the chamber [2].

During the cooling phase, the shape deviation between molding tools and molded glass occurs due to a CTE (coefficient of thermal expansion) difference [3]. Therefore, the quality of the molded lens not only depends on the parameters used during the PGM process but also on the material and manufacturing quality of the molds. An iterative production chain is usually conducted to obtain the desired optical properties of the molded lens [4], as shown in the summary in Figure 1. This iterative chain due to the intolerant optical performance and tool surface quality is time- and resource-consuming but can be reduced by implementing numerical simulations to predict the final geometrical shape of the molded component. Therefore, a more efficient tool design and manufacturing can be achieved [5]. For instance, Dambon et al. developed an efficient mold manufacturing process by incorporating shrinkage error prediction into a simulation model, which allows for mold manufacturing, starting directly with a compensated design [5].

Besides shape deviations, further outcomes can be considered in the FEM simulation. Zhao et al. found that, in PGM, an inhomogeneous temperature distribution occurs inside the molded lenses during rapid cooling, resulting in an induced refractive index drop [6], which is due to a higher cooling rate that is different from that during glass manufacturing. Su et al. was able to integrate a compensation procedure into the FEM model to predict the form deviation as well as the refractive index variation [7]. Additionally, glass characteristics are fundamental and should be considered as an input in the FEM simulation to accurately predict the molded lenses’ shape. For instance, Yi and Jain compared the final shape and volume of molded lenses with simulated data by incorporating the stress and structure relaxation characteristics of the glass to the FEM model. It was shown that the structure relaxation phenomenon of the glass has a substantial influence on the final lens shape, and it needs to be considered in the simulation model [8,9].

The versatility of this process enables the production of lenses made of different glass types. Chalcogenide glass (ChG) is a widely used material in industry and research facilities due to its excellent infrared optical properties [10]. The first PGM trials were performed by Zhang et al. in 2003, where the shape error of the molded lenses was approximately 0.5 µm in a 36 mm diameter spherical lens, reached after conducting several iteration loops with the metrology [11,12].

Therefore, the aim of this study is to integrate the relaxation behaviour of glass into a developed FEM simulation model in order to predict the refractive index variation and its distribution in molded lenses. A characterization method for the structural relaxation was developed and used as an input for the simulation model, which was validated by comparing the results with the measured surface deviation at different cooling rates.

## 2. Materials and Methods

### 2.1. Chalcogenide Glass as an Infrared Optic Material

In this study, an infrared-transmissive ChG glass produced by the company Schott AG (IRG 26, Mainz, Germany) was investigated. This glass material is widely used in infrared imaging applications due to its transmission in mid-wavelength infrared (3–8 µm) and part of the long-wavelength infrared (8–15 µm). The physical properties of this glass are published in the glass datasheet, as well as its Abbé number in the literature [13], as summarized in Table 1. Its Young’s modulus, listed in Table 1, exhibits low rigidity, even if compared with other ChG glasses [13]. In general, it is commonly described as a soft optical material that is very sensitive to surface defects [13]. Therefore, a ChG preform with a low surface quality will lead to a crack initiation during the molding process.

### 2.2. PGM Machine and Tools

A commercial precision glass molding machine, a Toshiba GMP-207HV—Numazu, Japan, was used for the molding experiments. It uses an infrared lamp to heat up the system, while cooling is achieved by opening the nitrogen valve to the chamber. The main system consists of mold dies, mold inserts, and cooling plates, where the mold inserts interact directly with the glass on the optical surface during molding, as shown in Figure 2. Therefore, a thin coating is applied to increase the service life of the inserts, enhance the wear resistance, and prevent glass adhesion [15]. In this work, a thin layer of CrAlN was deposited by means of PVD (Physical Vapor Deposition) technology on the tungsten carbide (CTN01L, Ceratizit S.A., Marmo, Luxembourg) mold inserts.

### 2.3. Material Characterization

Glass relaxation characteristics, namely, the structure and stress relaxation, should be experimentally determined by conducting self-developed dilatometric and molding experiments before they are implemented into the simulation model. Each of them is represented by a classic mathematical model, and the material-dependent parameters involved in the model must be characterized.

#### 2.3.1. Structure Relaxation and TNM (Tool–Narayanaswamy–Moynihan) Model

A glass at near- or below-transition temperatures (*T_g_*) has a structure, volume, and entropy that differ from the extrapolated equilibrium in a liquid state [16], as shown in Figure 3. This structure will evolve over time to a new equilibrium state at a new temperature. Hence, the structural relaxation occurs when the material is in a thermodynamic non-equilibrium state.

The TNM model is an analytical model that combines the contributions of three researchers: Tool [17], Narayanaswamy [18], and Moynihan [19]. This classical model has been verified by comparison with experimental results that agree well with the predictions of the model, and it is widely used in the field involving structural relaxation. [20,21].

Tool introduced an important concept called fictive temperature (*T_f_*) in his work, which became a cornerstone of the TNM model. Fictive temperature is defined as the actual equivalent temperature of an equilibrium state which corresponds to the given non-equilibrium state [22]. As shown in Figure 4, when a temperature down-jump from *T*_0_ to *T* is applied in Δt  instantaneously, the *T_f_* is still equal to the temperature *T*_0_. During the isothermal holding of the glass at the given T, structural changes take place, so the length changes gradually from L_0_ towards its equilibrium value *L*_∞_. When the structure relaxes completely, the *T_f_* is equal to the new equilibrium temperature *T*. A graphical definition of fictive temperature is the temperature value of the crossover point between the equilibrium line and an auxiliary line with slope $\alpha_{g}$ starting from the current point in the diagram length versus temperature, as shown in Figure 4, where *L*_0_ corresponds to the length immediately after the temperature down-jump. The blue points in the figure represent how the fictive temperature varies during the structural relaxation caused by a temperature down-jump.

To implement the TNM model into FEM analysis, Markovsky developed an efficient algorithm to calculate the fictive temperature in a numerical method [23], which is utilized in this work and is defined by the following equations.
(1a)$$T_{fi}\left(t\right)=\frac{T_{fi}\left(t-\Delta t\right)+T\left(t\right)\;\frac{\Delta t}{\tau_{i}}}{1+\frac{\Delta t}{\tau_{i}}}$$
(1b)$$\tau_{i}=\tau_{i,\;ref}\exp\left[-\frac{\Delta H}{R}\left(\frac{1}{T_{ref}}-\frac{x}{T\left(t\right)}-\frac{1-x}{T_{f}\left(t-\Delta t\right)}\right)\right]$$
(1c)$$T_{f}\left(t\right)=\overset{N}{\underset{i=1}{\sum }}g_{i}T_{fi}\left(t\right)$$
(1d)$$L\left(\;T\;\right)=L_{eq}\left(\;T_{0}\right)+\alpha_{l}\left(\;T_{f\;}-\;T_{0}\right)+\alpha_{g}\left(\;T\;-\;T_{f}\right)$$
where,
*R*:Ideal gas constant$\tau_{i,\;ref}$:Relaxation spectrum under the reference temperature$\tau_{i}$:Relaxation spectrumΔ*H*:Activation enthalpy*x*:Constant, 0 ≤ x ≤ 1$T_{fi}$:Fictive temperature spectrum$T_{f}$:Fictive temperature*T*:Temperature$g_{i}$:Weight factor for the fictive temperature spectrum*L_eq_*:Length in the equilibrium*α_g_*:CTE of solid glass*α_l_*:CTE of liquid glassΔ*t*:Time increment

A dilatometer DIL L75 PT Horizontal from the company Linseis GmbH—Selb, Germany was used to measure the CTE due to its contribution to the structure relaxation by Equation (1d). This device measures the sample length change under the contact driven by the pushrod. The contact force F applied can vary from 0.01 N to 5 N, depending on the application, and the maximal sampling frequency is 10 Hz.

The unavoidable contact force from the pushrod causes strong creep when measuring the glass expansion in the liquid state, as represented by the length decrease in high temperature (Figure 3). Therefore, a new setup was designed and implemented to measure the volumetric instead of the linear thermal expansion, where the glass was placed in an enclosed space, as shown in Figure 5.

The represented design was based on an investigation initially developed by Gottsmann et al. [24] and later supported by Yu et al. [25]. A constant contact force of 0.3 N was applied to the glass sample during heating, with a constant rate of 3 K/min up to 230 °C. However, the pushrod of the dilatometer measures the total displacement of glass and accessories (piston and base). This last one should be deducted to consider only the thermal expansion of the glass.

Regarding the variables in the equation group (1) used in the TNM model, a dilatometric temperature multi-jump experiment was carried out, in which a repeated heating process was avoided by integrating several jump experiments into one, as shown in the temperature profile curve in Figure 6.

#### 2.3.2. Stress Relaxation and General Maxwell Model

Stress relaxation is a phenomenon in which the stress decreases continuously, even though the strain is kept constant as $\varepsilon_{0}$, after which it was instantaneously applied at a moment t_1_, as shown in Figure 7a. The best-fit model to describe the stress relaxation is the generalized Maxwell model built by a parallel connection of several single Maxwell models which comprises a linear elastic spring and a serial connected viscous dashpot, as illustrated in Figure 7b [5].

When plotting the stress relaxation curves under different temperatures in a logarithm of time, the same shape is observed, and the curves seem to only be shifted by a distance. Thus, a stress relaxation curve at a certain temperature can be predicted through shifting a master curve under a reference temperature by a certain distance along the time axis; this kind of characteristic is called thermo-rheological simplicity (TRS) [26].

The stress relaxation can be measured by a four-point bending test, according to ASTM Standard C1116 (Figure 8a), where the specimen is clamped, while a compressive force is applied to it until it fails. As a result, the flexural modulus and bending stress are determined [27]. This setup was modified so the edge defects of the glass do not have an influence on the flexural strength, as depicted in the ring-on-ring test in Figure 8b [28]. In this work, the ring-on-ring setup is employed to determine the stress relaxation of the selected glass IRG 26. During the test, a stable control system is required, which can allow the tools and glass sample to reach the same desired temperature.

During the ring-on-ring test, the designed molds are installed in the precision molding machine GMP-207HV, and the system is heated up to a defined temperature, which is usually above the glass transition temperature (*T_g_*). The lower tool moves rapidly upwards to generate a step-jump of strain excitation to the glass sample. This position is then held during the entire relaxation process. The reaction force response is recorded by an external device with a high sampling rate of 10 kHz. After conducting a set of ring-on-ring tests under different temperatures (193 °C, 200 °C, 206 °C, and 211 °C), the TRS property can be fitted by the Williams, Landel, and Ferry (WLF) equation [29].

### 2.4. Simulation Model

The finite element simulation was built in a 2D axisymmetric thermomechanical model. Therefore, the viscoelastic behaviour of the glass and the elastic behaviour of the molding tools are considered. The thermal aspects such as radiation, convection, and conduction are also coupled in the simulation model. The commercial FEM software ABAQUS (2019, Paris, France) was used to define viscoelasticity and TRS behaviour for glass. To implement structure relaxation behaviour, we programmed predefined subroutines in Fortran to define the comprehensive incremental length change, called UEXPAN and SDVINI.

The key goal from the simulation is to predict the deformation and stress of the glass part during and after the molding process. Therefore, its mesh is nine times denser than those applied for other components, such as mold inserts or dies, as shown in Figure 9. Additionally, between glass and inserts, as well as between glass and lower dies, a friction behaviour is defined. Thermal contact conductance between adjacent parts is also considered, so heating and cooling of the glass can be carried out, since the lamp’s infrared light is transmitted directly through the IR glass, and it can be hardly heated up only by radiation. The simulation results include the mesh structure after the molding process and the predicted refractive index distribution in the mesh, according to the modified Lorentz–Lorenz equation [30], which considers the refractive index variation as the variation in the material density.

## 3. Results

### 3.1. Material Characterization

The results of the material characterization are used as an input for the simulation model. The measurement of the CTE of the IRG 26 glass in the solid and liquid states is represented in Figure 10. The linear CTE for the solid state is determined directly from the dilatometric experiments. The calculated value was 19.6 *×* 10^−6^ K^−1^, which is not significantly different from the value reported by the glass manufacturer [14]. For the liquid state, the setup and methodology described in Section 2.3.1 is applied. However, due to the application of the designed accessories, as shown in Figure 5, the slope (ω_L_) of the length variation (ΔL) on the liquid state region corresponds to the volumetric CTE and should be converted to the linear dimension. The calculated value for the linear CTE in the liquid state was 90.0 × 10^−6^ K^−1^.

For determining structure relaxation, the temperature multi-jump experiment was conducted using the dilatometer as well. However, the length response during the temperature jump experiment is composed of three components: the one related to the structural relaxation or configurational component, the one related to the rapid temperature change or vibrational component, and, finally, the one related to the creep or viscous component [31]. Therefore, only the configurational component is related to the structure relaxation under the corresponding isothermal temperature. The relaxation modulus (*M*_v_) is then calculable by applying Equation (2), as deducted by Narayanaswamy [18], where *p* are the properties involved in the structure relaxation, which is the length in this research, and *T*_2_ is the temperature after the temperature jump.
(2)$$M_{v}=\frac{p\left(T_{2},\;t\right)-p\left(T_{2},\;\infty \right)}{p\left(T_{2},\;0\right)-p\left(T_{2},\;\infty \right)}$$

On the other hand, it was found that the curve of the relaxation modulus could be described by the Kohlrausch–Williams–Watts (KWW) function in Equation (3) to stretch the exponential relationship assumed by Tool [26]:(3)$$M_{v}\left(\xi \right)=e^{-{\left(\xi \right)}^{\beta }}$$
(4)$$\xi \;=\overset{t}{\underset{0}{\int }}\frac{dt}{\tau_{\;}}$$
where
β:Stretching parameter, 0 < β ≤ 1ξ:Reduced time

The calculated relaxation moduli (*M*_v_) from the data in Equation (2) and the fitted Prony series under the five experimental temperatures are defined and plotted in Figure 11, where 175 °C was selected as the reference temperature.

The parameters defined in the TNM model are obtained by applying an optimization algorithm illustrated in Figure 12. The optimizer is called fmincon from MATLAB^®^ (R2019b, MA, USA), which is a constrained nonlinear optimization function for multi-variables.

The relaxation modulus (*M*_v_) can be calculated by the experimental data, including time, temperature, and the initial fictive temperature, which is exactly the temperature before the jump starts. On the other hand, the optimizer will iterate the parameter values for $\frac{\Delta H}{R},x,\beta ,\tau_{0}$. Following the TNM model, the theoretical relaxation modulus is obtained. When the difference between the experimental and theoretical relaxation modulus converges towards zero and the minimum is found, the parameters for the TNM model are obtained, as represented in Table 2, and used in the FEM simulation.

On the other hand, the stress relaxation curve under 193 °C from the ring-on-ring experiment was taken as the reference curve, as detailed in Section 2.3.2. The shift factors for the curves under other temperatures are fitted and plotted in Figure 13. The summary of the stress relaxation behaviour of IRG 26 glass is detailed in Table 3.

### 3.2. Validation of the Implemented Simulation Model

The simulation model including the structure and stress relaxation is validated through implementing real molding processes and comparing them with FEM results if the simulated molding process—specifically, the simulated temperature and force profiles—is close to the sensor values recorded. The flat molds illustrated in Figure 2 were used for the experimental procedure in which the IRG 26 glass was molded at 210 °C with a forming force of 3 kN and a holding force of 5 kN. In this work, a slow cooling rate of about 0.07 K/s, followed by a fast cooling stage of about 0.36 K/s, were applied, and it will be detailed in this chapter.

#### 3.2.1. Validation of Surface Form

During precision glass molding, the slow cooling stage and, then, the fast cooling stage have a more significant impact on the surface deviations than the heating and soaking stages. Therefore, a slow cooling rate was considered (0.07 K/s) for the process. The temperatures at the edge of the dies and in the center of the upper and lower molds were measured by thermocouples and plotted along with the simulated ones at the similar position in the FEM model, as shown in Figure 14.

It is noticed that the cooling temperature profile shows a maximum discrepancy of only 3 K for the upper mold, which even reduces with time. By combining the greater simulation error on the upper surface in Figure 14, this 3 K discrepancy is still critical to the prediction of the molded lens form. The surface validation is depicted in Figure 15, where the simulated lower surface fits the measurement with a deviation of 0.15 µm, while the upper side also shows a limited deviation of 0.4 µm.

The temperature distribution inside of the molded lens under the experimental temperature and force profile was simulated. Figure 16 shows the temperature distribution after controlled cooling (0.07 K/s). To provide a controlled cooling rate, the lamp was turned on during this stage, which can explain the simulated temperature distribution, where the exterior of the lens is not the cold area, and a homogenous temperature distribution (inhomogeneity less than 3 K) is predicted. Meanwhile, during the fast cooling stage, the lamp was turned off, so the exterior of the lens gets colder than it is in its center region.

#### 3.2.2. Simulation of Inhomogeneity

The OPD, as a result of interpreting the homogeneity of the refractive index in the molded lens, can be predicted through the simulation model combined with the ray-tracing method. The rays enter the glass mesh structure perpendicularly, and their trajectories through the mesh structure are traced by Snell’s law under the predicted refractive index distribution according to the Lorentz–Lorenz equation, as shown in Figure 17, where the passed elements are marked in red. Therefore, the OPD can be simulated at different radii, taking the refractive index variation and the surface deviation into account.

The OPD, due to surface deviation, can be removed by applying another ray-tracing calculation, where the overall refractive index is assigned as the nominal value from the datasheet. The OPD resulting from the inhomogeneity is then obtained by the difference between the two calculated OPDs, as shown in Figure 18, along with the predicted refractive index distribution. The range of refractive index variation is 8.0 × 10^−4^ (Figure 18b). Additionally, the average refractive index is 2.778, which means a dropping of around 0.75% of the nominal refractive index (2.799 at a wavelength of 3.4 µm).

## 4. Conclusions

The conducted investigation represents one of the first holistic FEM simulation models featuring shape deviations and bulk material alterations. It was successfully developed, implemented, and validated by considering the structure relaxation of the glass, which will improve the quality prediction for molded IRG 26 lenses. Furthermore, this work shows an innovative approach to determining CTE in the liquid state by using a dilatometer and a sleeve piston set-up that simplifies this measurement. The same device was used to determine the structure relaxation behaviour of the glass by applying a multi- jump experiment method, which enables a time-efficient solution. For the generalized Maxwell model, a ring-on-ring experiment was carried out under different temperatures in a glass molding machine. The Prony series of the normalized stress relaxation curve under a reference temperature were fitted, as well as the parameters in the WLF shift function. The material characteristics obtained from the experimental effort were used as inputs for the FEM model.

The glass properties were transferred into the material definition of the simulation model on the commercial simulation platform ABAQUS. To validate the prediction from the simulation results, the molding process was modelled to keep the simulated temperature and force profiles close to the values read from the sensors during the practical molding process. Furthermore, the predicted OPD resulting from the inhomogeneity was conducted by applying a ray-tracing method based on the simulation results.

This model was validated by a series of experiments and measurements, as shown in Figure 19. It was seen that the accuracy of the prediction of the form deviation highly depends on the structure relaxation behaviour during PGM. The error on the P-V (Peak to Valley) form deviation compared to the measurement is around 6.3 µm for the simulation conducted without structure relaxation, while for the one simulated with structure relaxation, it was 0.2 µm. Thus, this improved simulation model can simulate the molding process more accurately and reduce costly iterations in the lens production chain.

The presented results show the feasibility and advantage of this methodology, where the structure relaxation was integrated into the FEM simulation to predict the optical properties of molded lenses. In future work, the refractive index distribution with a variation of 8.0 × 10^−4^ and an average index drop of around 0.75% of the nominal refractive index should be validated.

## Figures and Tables

**Figure 1 materials-15-06756-f001:**
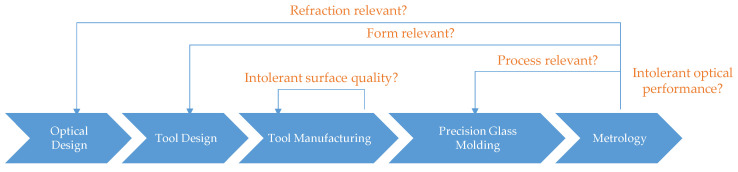
Iteration chain in the PGM production process.

**Figure 2 materials-15-06756-f002:**
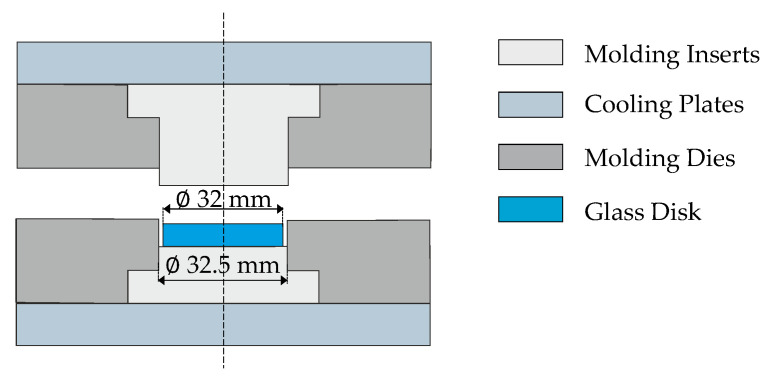
Conceptual view of components (closed-die concept).

**Figure 3 materials-15-06756-f003:**
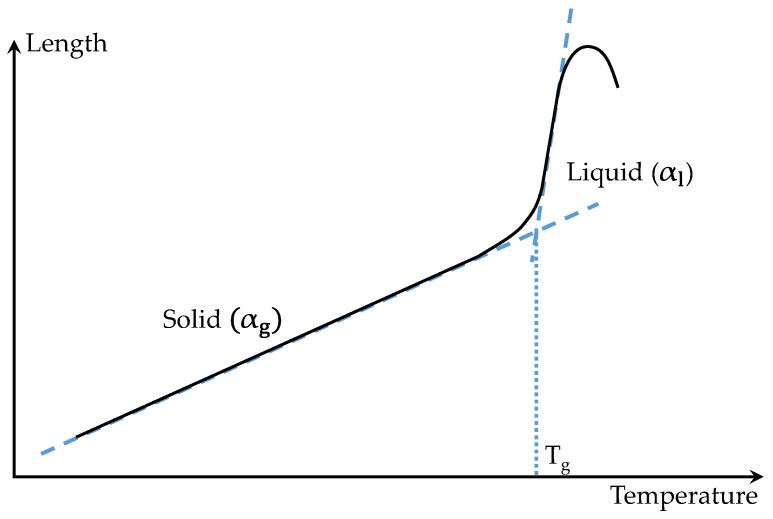
Plot of length change against temperature for a common glass.

**Figure 4 materials-15-06756-f004:**
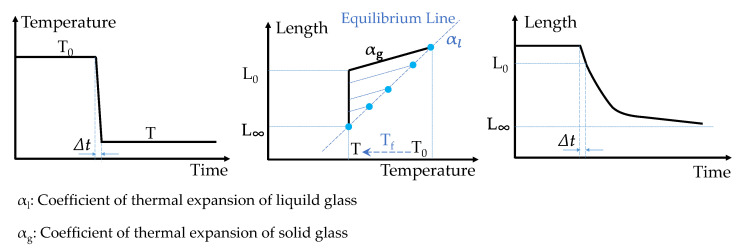
Schematic interpretation of fictive temperature in a temperature down-jump.

**Figure 5 materials-15-06756-f005:**
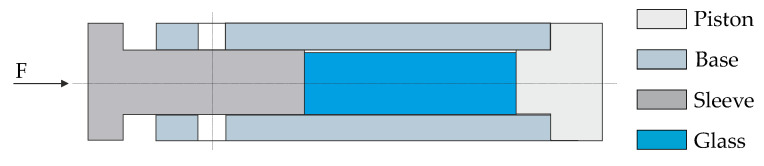
Measurement setup of the CTE of glass in a liquid state.

**Figure 6 materials-15-06756-f006:**
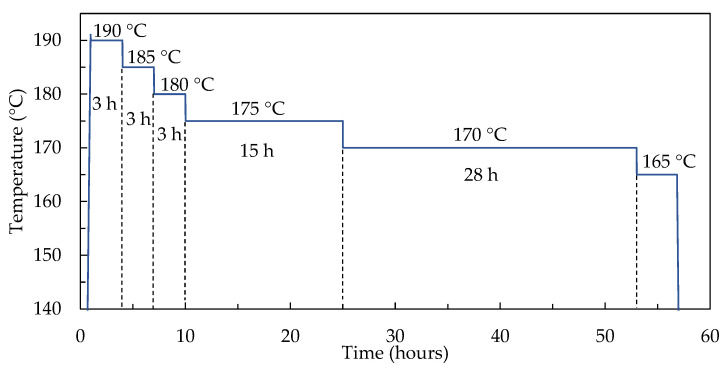
Details of the multi-jump experiment.

**Figure 7 materials-15-06756-f007:**
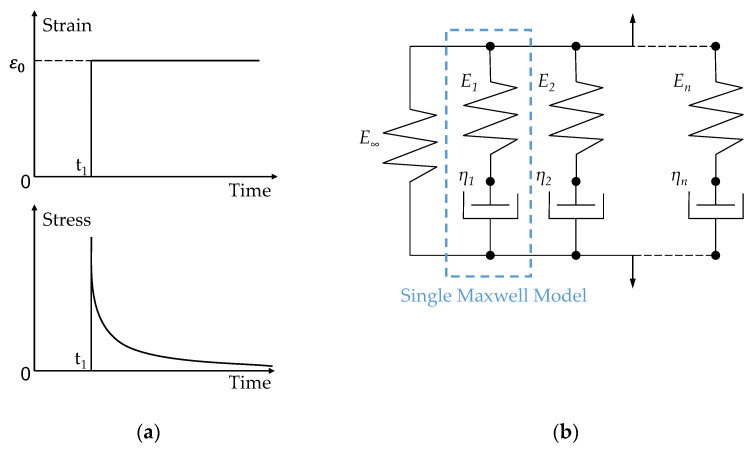
(**a**) Stress relaxation response to a strain in a step-jump excitation and (**b**) generalized Maxwell model (E: stiffness of the spring element; η: viscosity of the dashpot element).

**Figure 8 materials-15-06756-f008:**
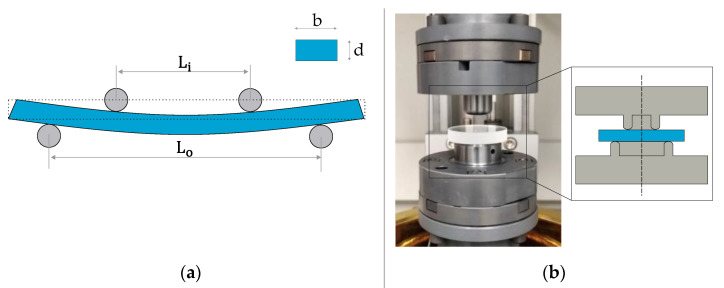
Four-point bending test (**a**) according to the ASTM Standard C1116. (**b**) Self-developed setup used for the ring-on-ring test.

**Figure 9 materials-15-06756-f009:**
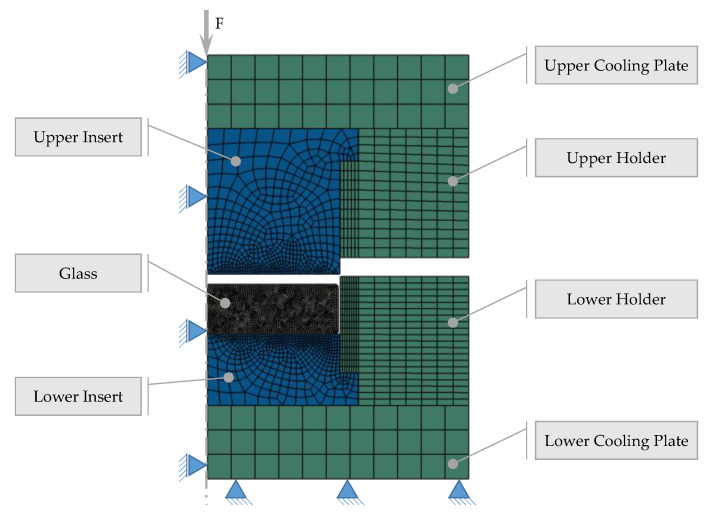
Overview of the simulation model; denser mesh for the glass part.

**Figure 10 materials-15-06756-f010:**
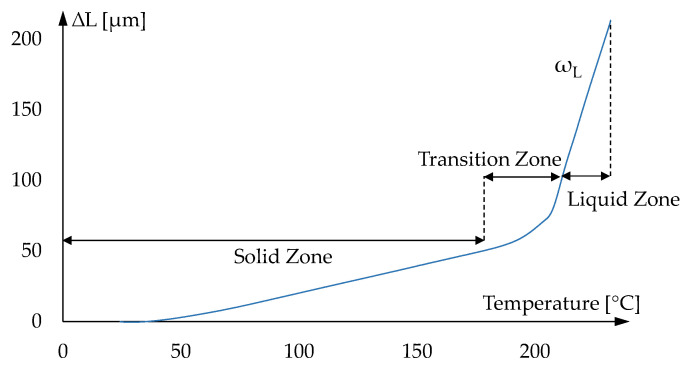
Coefficient of thermal expansion measurement of IRG 26 glass.

**Figure 11 materials-15-06756-f011:**
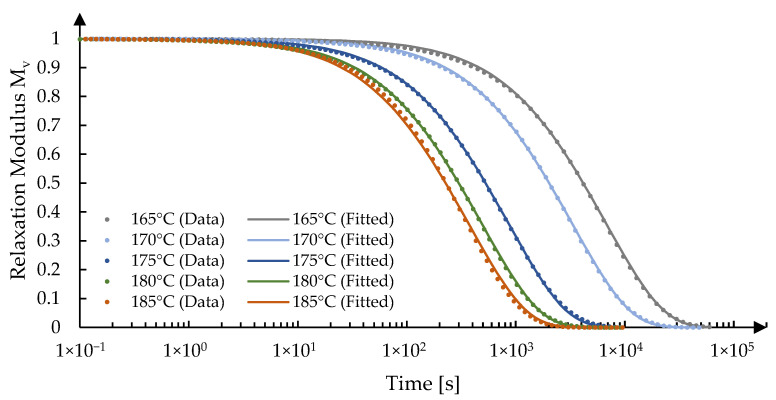
Measurements and fitting results of the structure relaxation modulus of IRG 26 Glass.

**Figure 12 materials-15-06756-f012:**
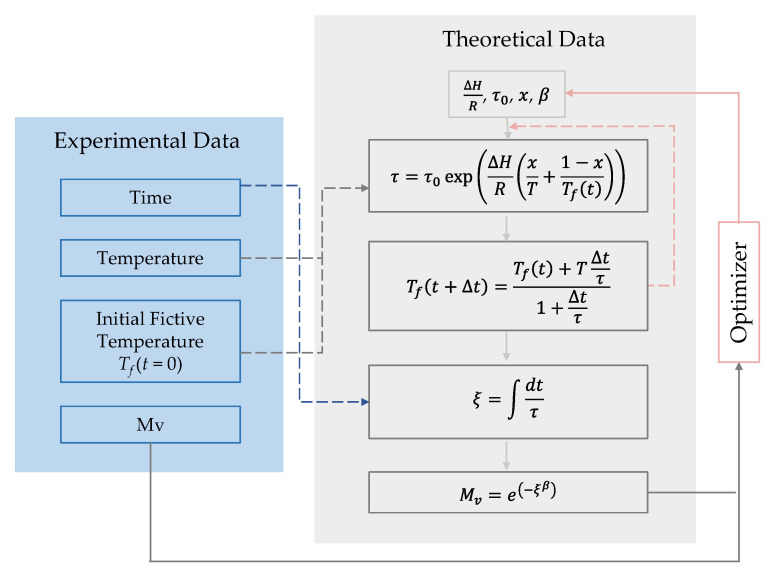
Illustration of the optimization algorithm.

**Figure 13 materials-15-06756-f013:**
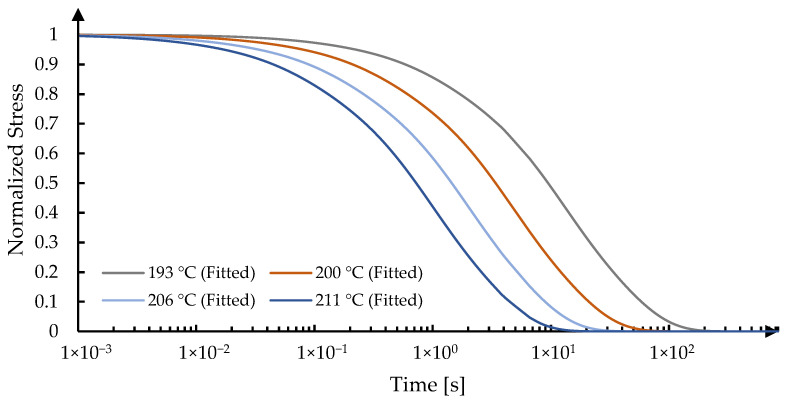
Fitting results of the stress relaxation behaviour of IRG 26 Glass.

**Figure 14 materials-15-06756-f014:**
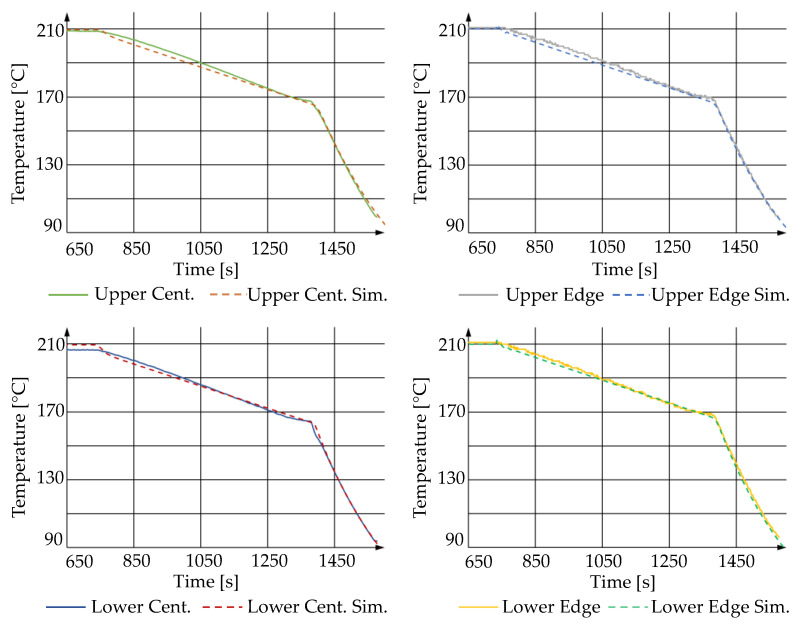
Gradual cooling profile comparison between real and simulated molding processes.

**Figure 15 materials-15-06756-f015:**
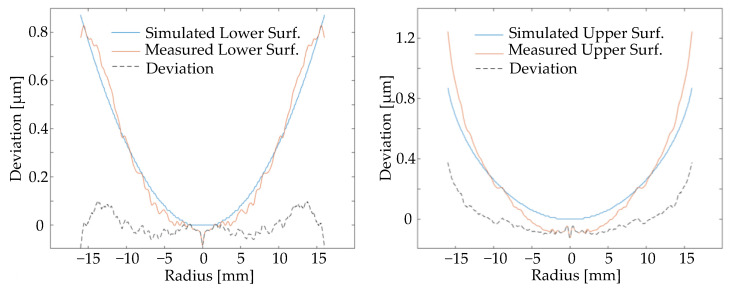
Surface form comparison between real and simulated molding processes.

**Figure 16 materials-15-06756-f016:**
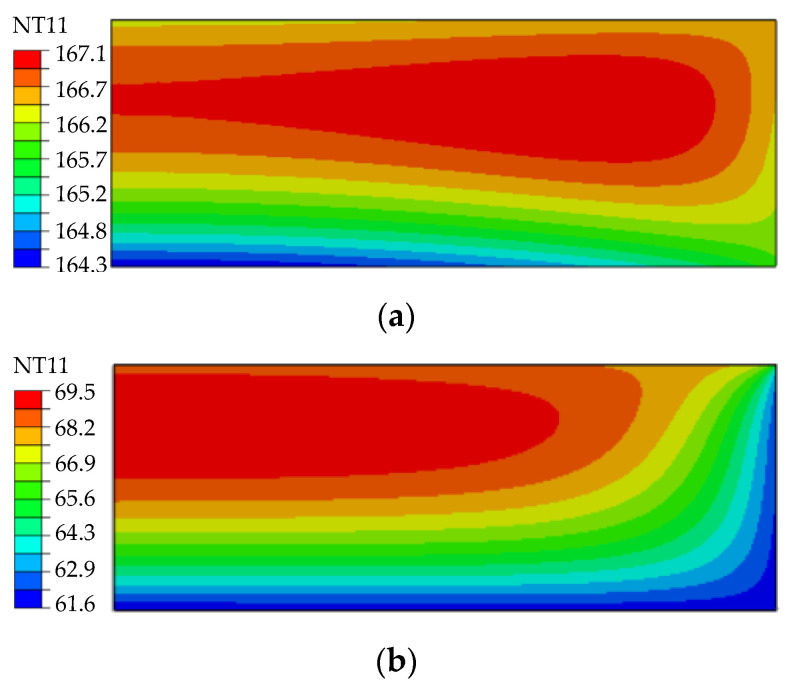
Simulated temperature distribution in molded glass: (**a**) after the controlled cooling stage; (**b**) after fast cooling. (NT11: Nodal temperature.)

**Figure 17 materials-15-06756-f017:**
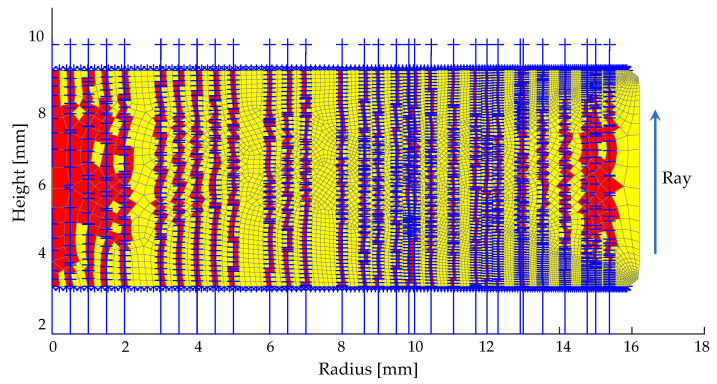
Meshed molded optics with OPD calculation based on ray-tracing.

**Figure 18 materials-15-06756-f018:**
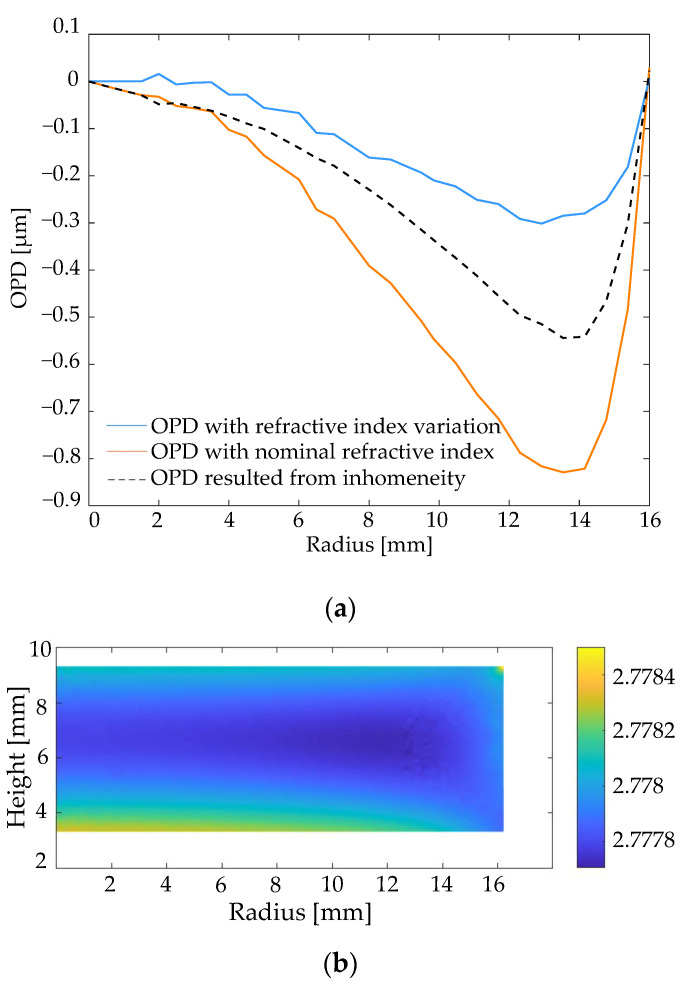
(**a**) OPD in the radial distance resulting from inhomogeneity. (**b**) Predicted refractive index distribution after the molding process.

**Figure 19 materials-15-06756-f019:**
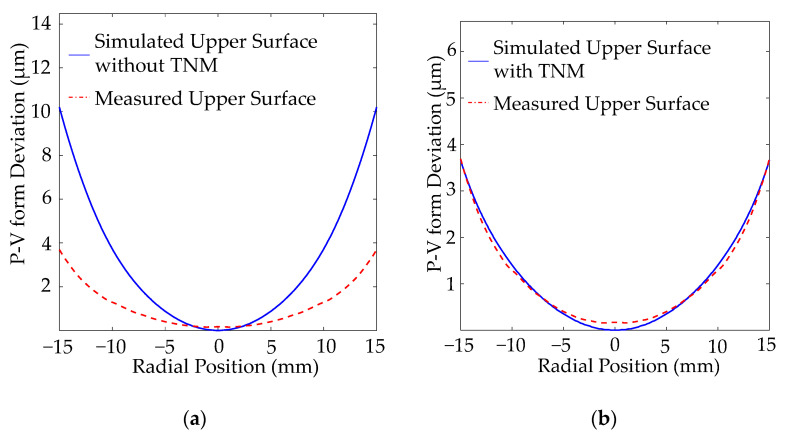
Validation of the FEM simulation model (**a**) without structure relaxation, (**b**) with structure relaxation.

**Table 1 materials-15-06756-t001:** Main material properties of IRG 26 glass [13,14].

Refractive Index (at 3 µm)	2.8015
Abbe Number (at 3–5 µm)	167
Glass Transition Temperature	185 °C
Density	4630 kg/m^3^
Young´s Modulus	18.3 GPa
Coefficient of Thermal Expansion (20–100 °C)	20.7 × 10^−6^/K
Thermal Conductivity	0.24 W/(m·K)

**Table 2 materials-15-06756-t002:** Structure relaxation parameters of the Prony series and shift function of glass IRG 26.

**Prony Series of Structure Relaxation**	**Parameter**	**Fitted Value**	**Parameter**	**Fitted Value**
	g_1_	0.0118	τ_1_	36.75
	g_2_	0.0328	τ_2_	100.0
	g_3_	0.2542	τ_3_	332.4
	g_4_	0.1585	τ_4_	754.8
	g_5_	0.5427	τ_5_	1556
**Shift Function**	**Parameter**		**Fitted Value**	
	x		0.68	
	∆H/R		38,392	

**Table 3 materials-15-06756-t003:** Stress relaxation parameters of the Prony series and WLF shift function of glass IRG 26.

**Prony Series of Stress Relaxation**	**Parameter**	**Fitted Value**	**Parameter**	**Fitted Value**
	G_1_	0.0719	τ_1_	0.0946
	G_2_	0.4040	τ_2_	0.7367
	G_3_	0.4244	τ_3_	4.7185
	G_4_	0.0997	τ_4_	29.959
**WLF Shift Function Parameters for TRS**	**Parameter**		**Fitted Value**	
	C_1_		32.67	
	C_2_		451	
	T_r_		178	

## Data Availability

Not applicable.

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
