# Peer review of "Simulation of the Refractive Index Variation and Validation of the Form Deviation in Precisely Molded Chalcogenide Glass Lenses (IRG 26) Considering the Stress and Structure Relaxation"

_materials, 2022, doi:10.3390/ma15196756_

Round 1
Reviewer 1 Report
The authors proposed a simulation process for precise infrared optics. They developed, implemented, and validated by considering the structural relaxation of the glass, which will improve the quality prediction for moulded IRG 26 lenses.
I recommend that this manuscript can be accepted for publication after revision. Some questions and suggested modifications are listed as follows:
1. Figure 1 is too simple. Please improve it and add details. And confirm whether Caption is wrong.
2. Why don't these graphs have corresponding values (Fig. 3,4,7)? This will make it difficult for the reader to understand; please add, if possible.
3. Should the variation of the thickness of the IRG 26 glass need to be considered in this work?
4. Did the authors study the optical performance of the refractive index variation and validation of the form deviation?
Author Response
- The content of the figure 1 is working as it's supposed to be, to show how are iterations carried out. I have added the criteria of the iterations into the figure. Other reviewers have mentioned also the caption, but on my side everything is normal. Please check if the error is still there in the revised version.
- These figures are used to help readers understand the theory and they are not from the experiments.
- The thickness compared to the refractive index change and the form deviation stands not in the focus center in this article where the planar molding is researched.
- Firstly the refractive index and form both belong to the optical property of a lens and can be obtained/simulated from the FEM simulation, which is exactly one of the outstanding points in our article. However, the validation for the refractive index is also a huge topic. As mentioned in the article we will try to reach this in the future work. And currently we can do the validation for the form, so we write this article with a reasonable title formulation.
Reviewer 2 Report
The submitted manuscript is entitled “Simulation of the Refractive Index Variation and Validation of the Form Deviation in Precisely Moulded Chalcogenide Glass Lenses (IRG 26) Considering the Stress and Structure Relaxation”.
Research on producing precise glass optics in an efficient manner is one of the important topics in material science and engineering. Generally, precision glass moulding (PGM) enables the production of an aspherical lens and irregular optical products in a single step. In the presented study, the authors investigate the key parameters that influence the PGM process. The authors utilized the finite element method (FEM) to simulate the thermomechanical process, predicting the optical properties of moulded chalcogenide lenses. The simulation was built in a 2D axisymmetric thermomechanical model with consideration of the viscoelastic behavior of the glass and the elastic of the moulding tools. The structure relaxation was integrated into the FEM simulation to predict the optical properties of the moulded lenses. The investigated material is an infrared-transmissive ChG glass (IRG 26), widely used in infrared imaging applications due to its transmission in mid-wavelength infrared and part of the long-wavelength infrared.
The manuscript requires some corrections in the presentation of the results and clarification of some issues. The authors should carefully review the manuscript and repair the problems.
In Section 2.3.2. Stress Relaxation and General Maxwell Model, the authors should introduce and briefly discuss quantities displayed in Figure 7, e.g. ε0, t1 etc.
The residual stresses in a lens depend mainly on the thermal history in the supercooled liquid as the effect of the variability and heterogeneity of thermal expansion. The stresses can be reduced by decreasing the cooling rate from the moulding temperature to the glass transition temperature. The authors should briefly comment on this.
In Section 3.2 Validation of the Implemented Simulation Model, this slow and fast cooling rate should be defined.
The authors should pay attention to the subscripts; e.g. in line 34, the subscript for Tg should be applied.
The multiplication sign is not necessary in Equations 1a and 1b.
According to the units widely used in the SI system, the time unit (second) is abbreviated as ‘s’ and not ‘sec’.
The value of some quantities and the tick labels in the figures should be displayed in the scientific form ‘10x’ instead of ‘Ex’.
Figure 3: The liquid region should be marked with the textbox (Liquid α1) at temperatures higher than Tg.
The caption of Figure 1: The typo in the reference should be repaired.
The displayed errors with the reference to Figures should be repaired.
Author Response
Thank you very much for your comments, I have adjusted and improved the paragraphs where you have suggested.
Regarding to the residual stress I would not like to add it into this work, because stress analysis is no the topic in our article but the significance of the structure relaxation model for the FEM simulation of the PGM process.
Other reviewers have mentioned the caption errors too, but honestly speaking, everything is normal on my side, I am also very confused. Please check if this problem still happens in the revised version.
Reviewer 3 Report
The authors report a simulation work of the refractive index variation and validation of the form deviation in precisely moulded chalcogenide glass lenses by considering the stress and structure relaxation. It provides good results and the theoretical background used to justify the refractive index variation is relevant solid. I will recommend its publication in Materials providing the authors can address the following comments.
1.The text coding should be improved to aviod "Error! Reference source not found."reports.
2.The display of the figures should be reorganized to 5-6 figures, in order to have a compact manuscript.
3. For the convenience of readers, some previous literatures on similar topics should be mentioned and cited properly.
Selective Direct Laser Writing of Pyrolytic Carbon Microelectrodes in Absorber-Modified SU-8.Micromachines 12(5):564,2021.
Holographic Resonant Laser Printing of Metasurfaces Using Plasmonic Template. ACS Photonics 5(5) ,2018. DOI: 10.1021/acsphotonics.7b01358
Author Response
Thank you very much for your time on the review!
Other reviewers have mentioned the caption problem too, but frankly speaking, everything works well on my computer. I am also confused how to solve this. Please check the revised version if this problem is still there.
The stress und structure relaxation theory is difficult for the common readers to understand, so I think these figures are useful to provide help. I am sorry that I cannot follow this advise.
Thanks for your suggested literatures, but I am sorry that I didn't see the joint point to integrate them into our article. But we can have cooperation in the future work if you meet any difficulties on the lens manufacture for your laser application.